# Distributional regression: CRPS-error bounds for model fitting, model selection and convex aggregation

**Clément Dombry***
Université de Franche-Comté,
CNRS, LmB (UMR 6623),
F-25000 Besançon, France.
clement.dombry@univ-fcomte.fr

**Ahmed Zaoui**
Université de Franche-Comté,
CNRS, LmB (UMR 6623),
F-25000 Besançon, France.
ahmed.zaoui@univ-fcomte.fr

## Abstract

Distributional regression aims at estimating the conditional distribution of a target variable given explanatory co-variates. It is a crucial tool for forecasting when a precise uncertainty quantification is required. A popular methodology consists in fitting a parametric model via empirical risk minimization where the risk is measured by the Continuous Rank Probability Score (CRPS). For independent and identically distributed observations, we provide a concentration result for the estimation error and an upper bound for its expectation. Furthermore, we consider model selection performed by minimization of the validation error and provide a concentration bound for the regret. A similar result is proved for convex aggregation of models. Finally, we show that our results may be applied to various models such as Ensemble Model Output Statistics (EMOS), distributional regression networks, distributional nearest neighbors or distributional random forests and we illustrate our findings on two data sets (QSAR aquatic toxicity and Airfoil self-noise).

## 1 Introduction

**Motivation and related literature.** We consider in this paper the distributional regression problem, where we want to estimate the conditional distribution of a target random variable $Y$ given explanatory co-variates $X$. We assume $(X, Y) \in \mathbb{R}^d \times \mathbb{R}$ and we let

$$F_x^*(y) = \mathbb{P}\left(Y \leq y \,|\, X = x\right). \tag{1}$$

be the conditional cumulative distribution functions (c.d.f.). Estimation is built on a training sample $\mathcal{D}_n = \{(X_i, Y_i), 1 \leq i \leq n\}$ of independent identically distributed (i.i.d.) copies of $(X, Y)$.

Let us emphasize that distributional regression is much more challenging than standard regression where only a point prediction for $Y$ given $X = x$ is provided which typically reduces to an estimation of the conditional expectation. The conditional distribution provides a full account for the variability of $Y$ given $X = x$ and distributional regression is therefore a crucial tool for forecasting when a precise uncertainty quantification is required.

Distributional regression is of relevance in various applied fields. To name a few, let us mention statistical post-processing of weather forecast (Matheson and Winkler, 1976; Gneiting et al., 2005), forecasting of wind gusts (Baran and Lerch, 2015), of solar irradiance (Schulz et al., 2021), of ICU length stays during COVID-19 pandemy (Henzi et al., 2021a), of breast cancer ODX score in oncology (Al Masry et al., 2024)...

The methodology relies on probabilistic forecast where the forecaster produces a predictive distribution for the quantity of interest (Gneiting and Katzfuss, 2014). Fitting a distributional regression

---

*clement.dombry@univ-fcomte.fr

38th Conference on Neural Information Processing Systems (NeurIPS 2024).

model typically involves the minimization of a proper scoring rule, the most popular one being the Continuous Ranked Probability Score (CRPS, Matheson and Winkler, 1976; Gneiting and Raftery, 2007). It compares the actual observation with the predictive distribution in a comprehensive manner. Many different models have been proposed for distributional regression among which: Analog similar to a nearest neighbor method (KNN, Toth, 1989), Ensemble Model Output Statistics similar to Gaussian heteroscedastic regression (EMOS, Gneiting and Raftery, 2007), Isotonic Distributional Regression (Henzi et al., 2021b) and more recent machine learning methods such as Distributional Regression Network (DRN, Rasp and Lerch, 2018) or Distributional Random Forest (DRF, Ćevid et al., 2022).

Despite these successful achievements in terms of methods and applications, a sound theory for distributional regression via scoring rule minimization is still missing. Recently, minimax rates of convergence with CRPS-error have been considered for distributional regression (Pic et al., 2023). The aim of this paper is to provide statistical learning guarantees for model fitting, model selection and convex aggregation based on CRPS minimization.

Model selection and convex aggregation are very important techniques in the field of statistics and machine learning and have been used very successfully for regression function estimation, see Tsybakov (2003); Bunea et al. (2007). The methods use two independent samples; the first sample, called training sample, is used to construct the initial estimators which are constituted as a dictionary (a collection of candidates). The second sample, called the validation sample, is used to aggregate them. Model selection enables the selection of the best candidate in the dictionary, while convex aggregation provides the optimal convex combination from these candidates.

**Contributions.** Our main results include a concentration bound for the theoretical risk when a parametric model is fitted via CRPS empirical risk minimization. We also consider model selection and model aggregation via CRPS minimization on a validation set and provide concentration bounds for the regret. Our results are first derived under sub-Gaussianity assumptions and then extended under weaker moment assumptions. We show that they apply to several popular models such as EMOS, DRN, KNN or DRF and provide a short illustration on two different datasets.

**Structure of the paper.** Section 2 first provides some background on distributional regression and proper scoring rules and then presents precisely the main methods and goals. The main results are stated in Section 3: an oracle inequality for the estimation error in model fitting (Theorem 1), and concentration bounds for the regret in model selection (Theorem 2) and model aggregation (Theorem 3). In Section 4, some specific models are introduced and the assumptions for our results to hold are checked. A short illustration on two different data set is provided in Section 5. Finally, an appendix contains all the proofs as well as some additional results.

## 2 Background on distributional regression and main goals

### 2.1 Probabilistic forecast and its evaluation with scoring rules

We first consider the simple setting of probabilistic forecast without co-variate where a future observation $Y$ is predicted by a probability distribution $F$, called predictive distribution. Proper scoring rules are used in order to compare the predictive distribution $F$ and the materializing observation $y$ which are objects of different nature. Let $\mathcal{P}_0 \subset \mathcal{P}(\mathbb{R})$ denote a subset of the set of all probability measures on $\mathbb{R}$, often identified with their c.d.f. A scoring rule[2] on $\mathcal{P}_0$ is a function $S \colon \mathcal{P}_0 \times \mathbb{R} \to [0, +\infty)$. The quantity $S(F, y)$ is interpreted as the error between the predictive distribution $F$ and the materializing observation $y$. The mean error when $Y$ has "true" distribution $G$ is denoted by

$$\bar{S}(F, G) = \mathbb{E}_{Y \sim G}[S(F, Y)].$$

The following notion of proper and strictly proper scoring rule is central in the theory.

**Definition 1.** *The scoring rule $S$ is said proper on $\mathcal{P}_0$ when*

$$\bar{S}(F, G) \geq \bar{S}(G, G), \quad \text{for all } F, G \in \mathcal{P}_0. \tag{2}$$

*It is said strictly proper when equality in Eq. (2) implies $F = G$.*

---

[2]For the sake of simplicity, we consider only the case of real-valued observation $Y$ and of non-negative scoring rule. A more general definition can be found in Gneiting and Raftery (2007).

Stated differently, the scoring rule $S$ is strictly proper on $\mathcal{P}_0$ when

$$\underset{F \in \mathcal{P}_0}{\arg \min} \, \bar{S}(F, G) = \{G\}, \quad \text{for all } G \in \mathcal{P}_0.$$

The interpretation is that, in order to minimize its mean error, the forecaster has to predict the "true" observation distribution $G$.

In this paper, we consider the Continuous Ranked Probability Score (CRPS, Matheson and Winkler 1976). This scoring rule is defined by the formula

$$S(F, y) = \int_{\mathbb{R}} \left( \mathbb{1}_{\{y \leq z\}} - F(z) \right)^2 \mathrm{d}z \tag{3}$$

for all $F$ finite absolute moment. Here the subset $\mathcal{P}_0 \subset \mathcal{P}(\mathbb{R})$ is the Wasserstein space

$$\mathcal{P}_1(\mathbb{R}) = \left\{ F \in \mathcal{P}(\mathbb{R}) \colon m_1(F) = \int_{\mathbb{R}} |y| F(\mathrm{d}y) < \infty \right\}$$

of probability measures on $\mathbb{R}$ with finite first moment. One can easily check from this definition that

$$\bar{S}(F, G) = \int_{\mathbb{R}} G(z) \left( 1 - G(z) \right) \mathrm{d}z + \int_{\mathbb{R}} \left( F(z) - G(z) \right)^2 \mathrm{d}z,$$

which implies

$$\bar{S}(F, G) - \bar{S}(G, G) = \int_{\mathbb{R}} \left( F(z) - G(z) \right)^2 \mathrm{d}z. \tag{4}$$

This quantity is nonnegative and vanishes if and only if $F = G$, ensuring that the CRPS is a strictly proper scoring rule. For discrete predictive distributions of the form $F = \sum_{i=1}^n w_i \delta_{y_i}$ (with $\delta_y$ the Dirac mass at $y$), the CRPS can be computed simply (Gneiting and Raftery, 2007) by

$$S(F, y) = \sum_{i=1}^n w_i |y_i - y| - \frac{1}{2} \sum_{i \neq j} w_i w_j |y_i - y_j|.$$

## 2.2 Model fitting, model selection and convex aggregation

We briefly present the methods and objectives that we address in this paper.

### 2.2.1 Theoretical risk in distributional regression

In a regression framework, we observe a sample $\mathcal{D}_n = \{(X_i, Y_i), \, 1 \leq i \leq n\}$ of independent copies of $(X, Y) \in \mathbb{R}^d \times \mathbb{R}$. Distributional regression aims at estimating the conditional distribution $Y | X = x$ characterized by its c.d.f. $F_x^*$ defined in Eq. (1). The marginal distribution of $X$ is denoted by $P_X$. The forecaster uses the training sample $\mathcal{D}_n$ and some algorithm to build a *functional* estimator $\hat{F}_n \colon x \mapsto \hat{F}_{n,x}$ of the map $F^* \colon x \mapsto F_x^*$. The accuracy of this estimator is here measured by its theoretical risk

$$\mathcal{R}(\hat{F}_n) = \mathbb{E}\left[ S(\hat{F}_{n,X}, Y) \right].$$

where expectation is taken with respect to the joint law of $(X, Y)$. This quantity can be seen as the counterpart of the mean squared error in point regression. The excess risk of $\hat{F}_n$ is defined as

$$\mathcal{R}(\hat{F}_n) - \mathcal{R}(F^*) = \mathbb{E}\left[ S(\hat{F}_{n,X}, Y) - S(F_X^*, Y) \right] = \mathbb{E}\left[ \bar{S}(\hat{F}_{n,X}, F_X^*) - \bar{S}(F_X^*, F_X^*) \right] \geq 0.$$

The nonnegativity is ensured by the fact that $S$ is a proper scoring rule according to Definition 1. If $S$ is strictly proper, the excess risk is equal to 0 if and only if $\hat{F}_{n,x} = F_x^*$ almost everywhere (with respect to $P_X$). For the CRPS, Eq. (4) implies that the excess risk can be rewritten as

$$\mathcal{R}(\hat{F}_n) - \mathcal{R}(F^*) = \mathbb{E}\left[ \int_{\mathbb{R}} \left| \hat{F}_{n,X}(u) - F_X^*(u) \right|^2 du \right].$$

### 2.2.2 Model fitting

Our first interest lies in model fitting by empirical risk minimization. Here we consider a parametric family $(F_\theta)_{\theta \in \Theta}$, $\Theta \subset \mathbb{R}^K$, where $F_\theta \colon x \in \mathbb{R}^d \mapsto F_{\theta,x} \in \mathcal{P}_0 \subset \mathcal{P}(\mathbb{R})$. The empirical risk associated with $F_\theta$ is computed on the training sample $\mathcal{D}_n$ by

$$\hat{\mathcal{R}}_n(F_\theta) = \frac{1}{n} \sum_{i=1}^{n} S(F_{\theta,X_i}, Y_i)$$

and is an empirical counterpart of the theoretical risk $\mathcal{R}(F_\theta)$. Empirical risk minimization consists in finding

$$\hat{\theta}_n = \arg\min_{\theta \in \Theta} \hat{\mathcal{R}}_n(F_\theta) \tag{5}$$

and proposing the estimator $F_{\hat{\theta}_n}$ which is thought as almost optimal within the family $(F_\theta)_{\theta \in \Theta}$. A classical decomposition of the excess risk of the corresponding estimator is given by

$$\mathcal{R}(F_{\hat{\theta}_n}) - \mathcal{R}(F^*) = \left( \mathcal{R}(F_{\hat{\theta}_n}) - \inf_{\theta \in \Theta} \mathcal{R}(F_\theta) \right) + \left( \inf_{\theta \in \Theta} \mathcal{R}(F_\theta) - \mathcal{R}(F^*) \right).$$

where the two terms are called the estimation error and the approximation error respectively. The approximation error is deterministic and depends on the ability of the family $(F_\theta)_{\theta \in \Theta}$ to approximate $F^*$. The estimation error is random as it depends on the training sample $\mathcal{D}_n$. Our first goal is the following:

**Goal 1:** provide non asymptotic estimates for the estimation error $\mathcal{R}(F_{\hat{\theta}_n}) - \inf_{\theta \in \Theta} \mathcal{R}(F_\theta)$.

### 2.2.3 Model selection and convex aggregation

Our second interest lies in model selection and convex aggregation via validation error minimization. Here we suppose that a validation sample $\mathcal{D}'_N = \{(X'_i, Y'_i), \quad 1 \le i \le N\}$ is available, which is assumed independent of the training sample $\mathcal{D}_n$.

**Model selection.** A common situation in machine learning is that we have $M$ algorithms at hand that are trained on $\mathcal{D}_n$, resulting in models $\hat{F}_n^1, \ldots, \hat{F}_n^M$. In order to select the best model, we compute the empirical risks on the validation sample

$$\hat{\mathcal{R}}'_N(\hat{F}_n^m) = \frac{1}{N} \sum_{i=1}^{N} S(\hat{F}_{n,X'_i}^m, Y'_i)$$

and select the model

$$\hat{m} = \arg\min_{1 \le m \le M} \hat{\mathcal{R}}'_N(\hat{F}_n^m).$$

An oracle having access to the theoretical risk would have selected

$$m^* = \arg\min_{1 \le m \le M} \mathcal{R}(\hat{F}_n^m),$$

leading to the definition of the regret

$$\mathcal{R}(\hat{F}_n^{\hat{m}}) - \mathcal{R}(\hat{F}_n^{m^*}) = \mathcal{R}(\hat{F}_n^{\hat{m}}) - \min_{1 \le m \le M} \mathcal{R}(\hat{F}_n^m).$$

**Goal 2:** provide non asymptotic estimates for the regret $\mathcal{R}(\hat{F}_n^{\hat{m}}) - \min_{1 \le m \le M} \mathcal{R}(\hat{F}_n^m)$.

**Convex aggregation.** We define the convex aggregation of models $\hat{F}_n^1, \ldots, \hat{F}_n^M$ with weights $\lambda_1, \ldots, \lambda_M$ by

$$\hat{F}_{n,x}^\lambda = \sum_{m=1}^{M} \lambda_m \hat{F}_{n,x}^m.$$

Here $\lambda = (\lambda_1, \ldots, \lambda_M)$ is an element of the simplex $\Lambda_M = \{\lambda \colon \lambda_m \ge 0, \sum_{1 \le m \le M} \lambda_m = 1\}$. The best weights for convex aggregation are obtained by minimization of the validation error, i.e.

$$\hat{\lambda} = \arg\min_{\lambda \in \Lambda_M} \hat{\mathcal{R}}'_N(\hat{F}_n^\lambda).$$

**Goal 3:** provide non asymptotic estimates for the regret $\mathcal{R}(\hat{F}_n^{\hat{\lambda}}) - \inf_{\lambda \in \Lambda_M} \mathcal{R}(\hat{F}_n^\lambda)$.

# 3 Main results

We present our main results for model fitting, model selection and convex aggregation in distributional regression. We first focus on concentration bound under sub-Gaussianity assumptions. In a second step, we extend our results under weaker moment assumptions. The definition of sub-Gaussian random variables and distributions is given in Appendix A as well as some useful concentration inequalities. All the proofs are postponed to Appendices B, C and D.

## 3.1 Estimation error in model fitting

We provide concentration results for the estimation error when fitting a parametric model via empirical CRPS-error minimization according to the framework described in Section 2.2.2. Our working assumptions are the following.

**Assumption 1.** *(sub-gaussianity)*

    *i)* *the variable $Y$ is $\beta_1$-sub-Gaussian;*

    *ii)* *there exists $\beta_2 > 0$ such that $m_1(F_{\theta,X})$ is $\beta_2$-sub-Gaussian for all $\theta \in \Theta$.*

Assumption 1-*i)* is classical in a regression setting (Györfi et al., 2002; Biau and Devroye, 2015) while Assumption 1-*ii)* characterizes the sub-Gaussian behavior of the absolute moment of $F_{\theta,X}$. Importantly, Assumption 1 implies that the variable $Z_\theta = S(F_{\theta,X})$ is $\sqrt{2(\beta_1^2 + \beta_2^2)}$-sub-Gaussian for all $\theta \in \Theta$, see Proposition 5 in the appendix.

Our second assumption requires compactness of the parameter space and Lipschitz continuity of the model $(F_\theta)_{\theta \in \Theta}$. We denote by $W_1$ the Wasserstein distance of order 1 on the space $\mathcal{P}_1(\mathbb{R})$.

**Assumption 2.** *(regularity) The parameter space $\Theta \subset \mathbb{R}^K$ is compact and there exists a constant $L > 0$ such that $W_1(F_{\theta_1,x}, F_{\theta_2,x}) \leq L\|\theta_1 - \theta_2\|$ for all $\theta_1, \theta_2 \in \Theta$, $x \in \mathbb{R}^d$.*

This assumption ensures the Lipschitz continuity of the empirical risk $\theta \mapsto \mathcal{R}_n(F_\theta)$, see Proposition 6 and, by compactness, the existence of $\hat{\theta}_n$, the ERM estimator defined in Eq. (5). We provide in Section 4 examples of popular models verifying the two assumptions.

Our main result provides a concentration bound on the estimation error. We let $R > 0$ be such that $\Theta$ is included in the ball centered at 0 and with radius $R$.

**Theorem 1.** *Under Assumptions 1- 2, for all $\delta \in (0,1)$, the estimation error satisfies, with probability at least $1 - \delta$,*

$$\mathcal{R}(F_{\hat{\theta}_n}) - \inf_{\theta \in \Theta} \mathcal{R}(F_\theta) \leq \sqrt{\frac{c_\beta \log(2n^K/\delta)}{n}} \tag{6}$$

*with $c_\beta = 64(\beta_1^2 + \beta_2^2)$ and provided that $n$ is large enough so that $n \log(2n^K/\delta) \geq (48LR)^2/c_\beta$.*

The inequality (6) is commonly known as an oracle inequality. The convergence rate depends on $\sqrt{\log(2n^K/\delta)/n}$ with $K$ the dimension of the parameter space and $n$ the sample size. A key point in the proof is the the combinatorial complexity of $\Theta$ in terms of $\epsilon$-net, see Devroye et al. (1996). A bound in expectation can easily be deduced from Theorem 1 and its proof.

**Corollary 1.** *Under Assumptions 1- 2,*

$$\mathbb{E}\Big[\mathcal{R}(F_{\hat{\theta}_n}) - \inf_{\theta \in \Theta} \mathcal{R}(F_\theta)\Big] \leq 2\sqrt{\frac{c_\beta \log(2n^K)}{n}}$$

*provided $n$ is large enough so that $n \log(2n^K) \geq (48LR)^2/c_\beta$.*

In particular, the ERM approach is weakly consistent when $n$ goes to infinity.

## 3.2 Estimation error in model selection and convex aggregation

We provide concentration results for the regret in model selection and convex aggregation. Recall that the methodology is based on minimization of the test error on a validation sample, see the precise framework in Section 2.2.3. We need the following assumption.

**Assumption 3.** *(sub-Gaussianity)*

*i) the variable $Y$ is $\beta_1$-sub-Gaussian;*

*ii) conditionally on $\mathcal{D}_n$, there exists $\beta_n = \beta(\mathcal{D}_n) > 0$ such that $m_1(\hat{F}_{n,X}^m)$ is $\beta_n$-sub-Gaussian for all $m = 1, \ldots, M$.*

We will see in Section 4 that for several popular models such as distributional nearest neighbors or distributional random forest, Assumption 3-*ii)* is satisfied with $\beta_n = \max_{1 \le i \le n} |Y_i|$ which is of order $\beta_1 \sqrt{\log n}$ (Vershynin, 2018, Exercise 2.5.8 p.25).

Under this assumption, a control of the regret in model selection is provided by the following theorem. Our results hold conditionally on the training set $\mathcal{D}_n$.

**Theorem 2.** *Under Assumption 3, for all $\delta \in (0,1)$, the regret in model selection satisfies*

$$\mathbb{P}\left( \mathcal{R}(\hat{F}_n^{\hat{m}}) - \min_{1 \le m \le M} \mathcal{R}(\hat{F}_n^m) \le 4\sqrt{c_n \log(2M/\delta)/N} \, \Big| \, \mathcal{D}_n \right) \ge 1 - \delta$$

*with $c_n = \beta_1^2 + \beta_n^2$. Furthermore,*

$$\mathbb{E}\left[ \mathcal{R}(\hat{F}_n^{\hat{m}}) - \min_{1 \le m \le M} \mathcal{R}(\hat{F}_n^m) \, \Big| \, \mathcal{D}_n \right] \le 8\sqrt{\frac{c_n \log(2M)}{N}}.$$

When selecting the hyperparameter $k$ in nearest neighbor distributional regression or `mtry` in distributional random forest, one has respectively $M = n$ and $M = d$ if all possible values are considered – see Section 4 for more details. Note that when the response variable $Y$ is bounded, then $c_n$ does not depend on $n$ and is constant. We now state a bound for the regret in convex aggregation.

**Theorem 3.** *Under Assumption 3, for all $\delta \in (0,1)$, the regret in convex aggregation satisfies*

$$\mathbb{P}\left( \mathcal{R}(\hat{F}_n^{\hat{\lambda}}) - \inf_{\lambda \in \Lambda} \mathcal{R}(\hat{F}_n^\lambda) \le 8\sqrt{c_n \log(2N^M/\delta)/N} \, \Big| \, \mathcal{D}_n \right) \ge 1 - \delta,$$

*with $c_n = \beta_1^2 + \beta_n^2$ provided $N$ is large enough so that $N \log(2N^M/\delta) \ge 48^2/c_n$. Furthermore,*

$$\mathbb{E}\left[ \mathcal{R}(\hat{F}_n^{\hat{\lambda}}) - \inf_{\lambda \in \Lambda} \mathcal{R}(\hat{F}_n^\lambda) \, \Big| \, \mathcal{D}_n \right] \le 2\sqrt{\frac{c_n \log(2N^M)}{N}}$$

*provided $N \log(2N^M) \ge 48^2/c_n$*

### 3.3 Beyond sub-gaussianity

The preceding results hold under a strong sub-Gaussianity. We next adapt our results to the following weaker moment condition.

**Assumption 4.** *There is $p \ge 2$ and $D > 0$ such that $\mathbb{E}[|Y|^p] \le D$ and $\mathbb{E}[|m_1(F_{\theta,X})|^p] \le D$ for all $\theta \in \Theta$*

We recall that, for $p \ge 1$, the $L^p$-norm of a random variable $Z$ is defined by $\|Z\|_{L^p} = \mathbb{E}[|Z^p|]^{1/p}$.

**Theorem 4.** *Under Assumptions 2 and 4, we have*

$$\left\| \mathcal{R}(F_{\hat{\theta}_n}) - \inf_{\theta \in \Theta} \mathcal{R}(F_\theta) \right\|_{L^p} \le C n^{-p/(2(p+K))}.$$

*with constant $C > 0$ depending only on $K, p, L, D, R$ and possibly made explicit from the proof. This implies the bound for the expected estimation error*

$$\mathbb{E}\left[ \mathcal{R}(F_{\hat{\theta}_n}) - \inf_{\theta \in \Theta} \mathcal{R}(F_\theta) \right] \le C n^{-p/(2(p+K))}.$$

For $p$ large, the rate of convergence tends to the parametric rate $n^{-1/2}$ obtained (up to a logarithmic factor) in the sub-Gaussian case . We propose additional results for model selection (Theorem 5) and convex aggregation (Theorem 6) which are, for the sake of brevity, postponed to Appendix D.2.

## 4   Examples and popular models for distributional regression

We present the most popular models for distributional regression for which we want to apply our results. The first two (EMOS and DRN) are parametric, while the last two (distributional $k$-NN and DRF) are fully non-parametric.

## 4.1 EMOS and distributional regression networks

**EMOS.** The EMOS model was designed by Gneiting et al. (2005) for the purpose of statistical post-processing of ensemble weather forecast. In this framework, the predictive distribution takes the form of a discrete distributions $m^{-1}\sum_{l=1}^{m}\delta_{y_l}$ with members $y_1,\ldots,y_m$ corresponding to different scenarios obtained from numerical weather predictions. Such forecast typically suffer from bias and underdispersion so that statistical post-processing is needed. The explanatory variable for distributional regression are Ensemble Member Output Statistics such as the ensemble mean $\bar{y}$ and ensemble variance $v_y^2$. In its simplest version, EMOS models the predictive distribution as a Gaussian distribution with parameters $m = \beta_0 + \beta_1\bar{y}$ and $\sigma^2 = \beta_0' + \beta_1'v_y^2$. This is a parametric model with $\theta = (\beta_0, \beta_1, \beta_0', \beta_1')$ in $\Theta = \mathbb{R}^2 \times (0,\infty)^2$. Minimum CRPS estimation is used for model fitting as described in 2.2.2. This simple yet successful method has encountered many generalizations (Scheuerer 2013; Scheuerer and Hamill 2015; Baran and Nemoda 2016) and we shall consider the following general setting. Given $x \in \mathbb{R}^d$, the predictive distribution takes the form $F(\cdot;\theta) = F((\cdot - m)/\sigma)$ with $\theta = (\alpha, \beta, \alpha', \beta') \in \mathbb{R}^{1+d} \times \mathbb{R}^{1+d}$ and

$$\begin{cases} m(x;\theta) = \alpha + \beta^\top x \\ \sigma^2(x;\theta) = \text{softplus}(\alpha' + {\beta'}^\top x) \end{cases}$$

with $\text{softplus}(u) = \log(1 + e^u)$. The predictive distribution belongs to a location/scale family and the location and log-scale parameters are linear in the covariate $x$. The $\text{softplus}$ link function ensures positivity of the scale.

**Distributional Regression Networks (DRN).** In order to consider higher dimension co-variates together with non-linear dependence, Rasp and Lerch (2018) introduce distributional regression networks. In a setting similar to EMOS, neural networks are used to model complex response functions $m(x;\theta)$ and $\log\sigma^2(x;\theta)$. In the case of a single hidden layer with $H$ units, the equations write

$$\begin{cases} m(x;\theta) = \alpha + \beta^\top g(\gamma + \delta^\top x) \\ \sigma^2(x;\theta) = \text{softplus}(\alpha' + {\beta'}^\top g(\gamma + {\delta'}^\top x)) \end{cases}, \tag{7}$$

where $g$ denotes the activation function acting componentwise, $(\alpha, \beta, \alpha', \beta') \in \mathbb{R}^{1+H} \times \mathbb{R}^{1+H}$ the parameters for the output layers and $(\gamma, \delta) \in \mathbb{R}^{H+Hd}$ the biases and weights of the hidden layer. The global parameter $\theta = \alpha, \beta, \alpha', \beta', \gamma, \delta)$ lies in dimension $(d + 3)H + 2$. Note that in absence of hidden layers, DRN reduces to EMOS. Extension to a MLP structure with multiple hidden layers is straightforward, see Schulz and Lerch (2022) for a review of DRNs and their applications.

We next provide conditions ensuring that our results hold for the EMOS and DRN models.

**Proposition 1.** *Let $(F_\theta)_{\theta\in\Theta}$ be a EMOS or DRN model with parameters restricted to a compact subset $\Theta$. If $Y$ is sub-Gaussian, $X$ is bounded and the activation function $g$ is Lipschitz continuous, then Assumptions 1, 2 and 3 are satisfied.*

## 4.2 Distributional k-Nearest Neighbors and Random Forests

**Distributional k-Nearest Neighbors (KNN).** The predictive distribution is built on a straightforward extension of $k$-nearest neighbor regression: we set

$$\hat{F}_{n,x}(y) = \frac{1}{k}\sum_{i=1}^{k}\mathbb{1}_{\{X_i\in\text{knn}(x)\}}\mathbb{1}_{\{Y_i\leq y\}}$$

where $\text{knn}(x)$ denotes the set of $k$ nearest neighbors of $x$ in the training set $(X_i)_{1\leq i\leq n}$. The main hyperparameter is the number $k$ of neighbors, leading to a model selection problem as considered in 2.2.3. This simple method is known as the Analog method in the framework of statistical post-processing of weather forecast (Toth, 1989).

**Distributional Random Forests (DRF).** Random forests for distributional regression have been introduced by Ćevid et al. (2022) and are a powerful nonparametric method. We describe only the main lines of the method. Recall that in standard regression, Breiman's Random Forest (Breiman,

2001) estimates the regression function by

$$\hat{\mu}(x) = \frac{1}{B} \sum_{b=1}^{B} T^{(b)}(x) = \frac{1}{B} \sum_{b=1}^{B} \frac{1}{|L^b(x)|} \sum_{i=1}^{n} Y_i \mathbb{1}_{\{X_i \in L^b(x)\}},$$

where $T_1, \ldots, T_B$ denote randomized regression trees built on bootstrap samples of the original data and $L^b(x)$ the leaf containing $x$ in the tree $T^b$. Interverting the two sums yields

$$\hat{\mu}(x) = \sum_{i=1}^{n} \left( \frac{1}{B} \sum_{b=1}^{B} \frac{\mathbb{1}_{\{X_i \in L^b(x)\}}}{|L^b(x)|} \right) Y_i = \sum_{i=1}^{n} w_{ni}(x) Y_i.$$

Using these *Random Forest weights*, the predictive distribution writes

$$\hat{F}_{n,x}(y) = \sum_{i=1}^{n} w_{ni}(x) \mathbb{1}_{\{Y_i \leq y\}}.$$

This strategy based on Breiman's regression tree is used by Meinshausen (2006) to construct the quantile regression forest. Different splitting strategies in the tree construction have been considered to better detect changes in distribution rather than changes in mean (Taillardat et al. 2016; Athey et al. 2019). Note that Ćevid et al. (2022) minimizes a scoring rule to construct the splits of the trees. The main hyperparameter of this nonparametric method is the so-called `mtry` parameter that controls the number of co-variates tested at each split in the trees.

For the distributional KNN and DRF models, model fitting with ERM is irrelevant and we only consider model selection and convex aggregation.

**Proposition 2.** *Let $\hat{F}_n$ be a KNN or DRF model fitted on a training set $\mathcal{D}_n$. Then Assumption 3-ii) is satisfied with $\beta_n = \max_{1 \leq i \leq n} |Y_i|$.*

## 5 Numerical results

In this section, we illustrate how model selection and model aggregation work on real data sets. The cross validation methodology, widely used in practice, is justified by the theoretical results obtained in Section 3. The source code for these experiments can be found at `https://github.com/ZaouiAmed/Neurips2024_DistributionalRegression`.

**Datasets.** We consider two datasets used in the framework of heteroscedastic regression with reject option as detailed in Zaoui et al. (2020). The first dataset, *QSAR aquatic toxicity* (Ballabio et al., 2019) is referred to as `qsar` and comprises 546 observations with 8 numerical features used to predict acute toxicity in Pimephales promelas. The toxicity output ranges from 0.12 to 10.05 with evidence of a low heteroscedasticity. The second dataset, *Airfoil Self-Noise* (Brooks et al., 2014) is referred to as `airfoil` and consists of 1503 observations with 5 measured features from aerodynamic and acoustic tests. The output represents the scaled sound pressure level in decibels, ranging from 103 to 140 with evidence of a strong heteroscedasticity..

**Models.** We consider the KNN and DRF models to predict the conditional distribution of the output variable. Our focus lies on hyperparameters selection via minimization of the validation error, where the main hyperparameter is the number k of neighbors and the number `mtry` of variables considered at each split for KNN and DRF respectively. We utilize the implementation of these methods from the R packages `KernelKnn` and `DRF`. For DRF, a preliminary exploration shows that sensible choices for the other parameters are `num.trees=` 1000, `sample.fraction=0.9`, `min.node.size=` 1 and default values for others parameters. Finally, the R package `ScoringRule` is used for CRPS computation and the `optim` function based on the Nelder-Mead method is used for parameter optimization in convex aggregation.

**Methodology.** We use the same methodology for the two datasets. We divide the data into three parts: 50% for training, 20% for validation and 30% for testing. In a first stage, the training set is used to train the model (KNN or DRF) for the various hyperparameters (k or `mtry`) and the validation set is used to select best hyperparameters $\widehat{k}$ or $\widehat{mtry}$. Furthermore, the validation set also used to

choose for model selection (MS), choosing between KNN and RF, and the best convex aggregation (CA). In a second stage, the different models (KNN, RF, MS and CA) are refitted on the union of training and validation sets (70% of data) and evaluated on the test set (30%) by computing the test CRPS-error. This process is repeated 100 times for different random splits of the data into training, validation and testing sets. The distributions of the test error for KNN, DRF, MS and CA is then analyzed in terms of mean, standard error and boxplot.

**Results.** For the sake of brevity, only the results for the `qsar` dataset are presented, while the results for the `airfoil` dataset are postponed to Appendix F. The first stage of the procedure where the hyperparameter (`k` or `mtry`) is selected by minimization of the validation error is shown on the left plot of Fig. 1 for KNN and on the middle plot for DRF. In both cases, the validation error curve shows a clear minimum allowing to select the hyperparameter $\widehat{k} = 8$ and $\widehat{\text{mtry}} = 4$ corresponding to CRPS-error of $0.696$ and $0.678$ on the validation set respectively. In the second stage, KNN, DRF, MS and CA are evaluated on the test set and the right plot of Fig. 1 shows the distribution of the test error over 100 repetitions. We can see from the boxplots that the distribution of the test error is slightly larger for KNN than for DRF, that MS achieves almost the same performance as RF and that CA achieves slightly better performance than DRF. Hence model selection and convex aggregation accomplish their goal. The numerical values of the means together with standard errors are summarized in Table 1, confirming our analysis from the boxplots.

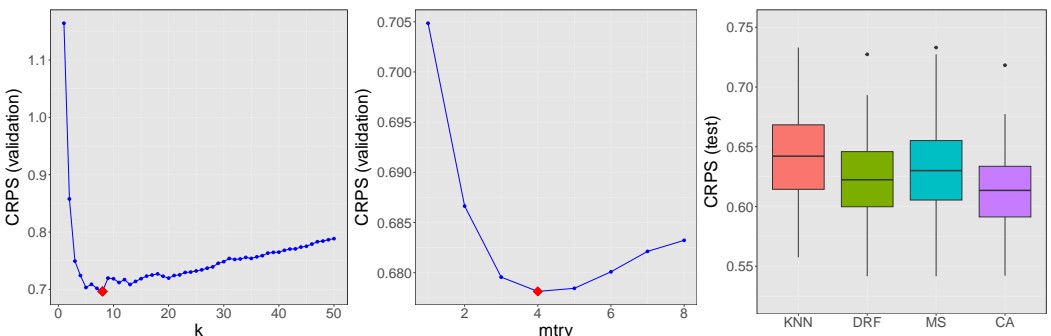

Figure 1: `qsar` data. Left and middle: selection of `k` for the KNN algorithm and of `mtry` for the DRF algorithm by minimization of the validation error. Right: test error evaluated with 100 repetitions for KNN, DRF, model selection (MS) and convex aggregation (CA).

| KNN | DRF | MS | CA |
|---|---|---|---|
| 0.643 (0.004) | 0.624 (0.004) | 0.632 (0.004) | 0.612 (0.003) |

Table 1: `qsar` data. Mean of the test CRPS and its standard error (in parenthesis) over 100 repetitions.

## 6   Conclusion

In this work, we considered distributional regression with error assessed using the CRPS and investigated model fitting via empirical risk minimization. Additionally, we explored classical aggregation procedures: model selection and convex aggregation where two independent samples are used, the first for model construction, the second for model selection or convex aggregation. We derived oracle concentration inequalities for the estimation error and established an upper bound for its expectation within the sub-Gaussian framework and beyond, considering weaker moment assumptions. These new theoretical results are solid mathematical justifications for common practices in the framework of distributional regression. In future work, we will study the minimax convergence rates for our approaches and consider the use of empirical process theory to strengthen our results.

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

# A  Sub-gaussian distributions and concentration inequalities

The concept of sub-Gaussianity has been extensively studied and is a key tool in deriving concentration inequalities, as documented in the monographs by Vershynin (2018) and Boucheron et al. (2013). For the convenience of the reader, we recall the definition and basic properties of sub-Gaussian random variables as well as the useful Hoeffding inequality.

## A.1  Sub-Gaussian random variables

**Definition 2** (Sub-Gaussian random variable). *Let $\beta > 0$. The random variable $X$ is said to be $\beta$-sub-Gaussian if*

$$\mathbb{E}\left[e^{\lambda(X-\mathbb{E}[X])}\right] \leq e^{\frac{1}{2}\lambda^2\beta^2} \quad \text{for all } \lambda \in \mathbb{R}.$$

**Proposition 3.** *Let $\beta, \beta_1, \beta_2 > 0$.*

> *i) If $X$ is $\beta$-sub-Gaussian, then $|X|$ is also $\beta$-sub-Gaussian.*
>
> *ii) If $X$ is a bounded random variable such that $X \in [a, b]$ almost surely, then $X$ is $\beta$-sub-Gaussian with $\beta = (b-a)/2$.*
>
> *iii) If $X, Y$ are random variables such that $X$ is $\beta$-sub-Gaussian and $0 \leq Y \leq X$, then $Y$ is also $\beta$-sub-Gaussian.*
>
> *iv) If $X_i$ is $\beta_i$-sub-Gaussian for $i = 1, 2$, the sum $Y = X_1 + X_2$ is $\beta$-sub-Gaussian with $\beta = \sqrt{2(\beta_1^2 + \beta_2^2)}$.*

## A.2  Concentration inequalities

In this section we gather several technical results which are used to derive the contributions of this work. We start with the general Hoeffding's inequality (Vershynin, 2018, Theorem 2.6.2 p. 28).

**Proposition 4** ( Hoeffding (1963)). *Let $Z_1, \ldots, Z_n$ be independent random variables such that $Z_i$ is $\beta$-sub-Gaussian for all $i = 1 \ldots, n$. Then, for all $t \geq 0$ we have*

$$\mathbb{P}\left(\left|\frac{1}{n}\sum_{i=1}^{n}\left(Z_i - \mathbb{E}[Z_i]\right)\right| \geq t\right) \leq 2\exp\left(-\frac{nt^2}{2\beta^2}\right).$$

The following Lemma is often useful to derive upper bound in expectation from concentration inequalities.

**Lemma 1.** *Let $a \geq 1$, $b > 0$ and $Z$ a positive random variable such that*

$$\mathbb{P}\left(Z \geq t\right) \leq \exp(a - bt^2) \quad \text{for all } t \geq 0.$$

*Then we have*

$$\mathbb{E}[Z] \leq \left(1 + \frac{\sqrt{\pi}}{2}\right)\sqrt{\frac{a}{b}} \leq 2\sqrt{\frac{a}{b}}.$$

*Proof.* The assumption implies that $\mathbb{P}\left(Z \geq t\right) \leq \min\left(1, \exp(a - bt^2)\right)$ for all $t \geq 0$. Then,

$$\mathbb{E}[Z] = \int_0^{+\infty} \mathbb{P}(Z \geq t)dt = \sqrt{\frac{a}{b}} + \int_{\sqrt{a/b}}^{+\infty} \exp\left(-(bt^2 - a)\right)dt. \tag{8}$$

Since $(u - v)^2 \leq u^2 - v^2$ for $0 \leq v \leq u$, we have

$$\int_{\sqrt{a/b}}^{+\infty} \exp\left(-(bt^2 - a)\right)dt \leq \int_{\sqrt{a/b}}^{+\infty} \exp\left(-b\left(t - \sqrt{a/b}\right)^2\right)dt = \frac{\sqrt{\pi}}{2\sqrt{b}} \leq \frac{\sqrt{\pi}}{2}\sqrt{\frac{a}{b}}, \tag{9}$$

where the last inequality uses $a \geq 1$. Combining Eq. (8) and Eq. (9) yields the result. $\qquad\square$

### A.3 Application to the empirical risk

The results from the last two sections are applied to the CRPS and the empirical risk.

**Proposition 5.** *Under Assumption 1, for all $\theta \in \Theta$, the random variable $S(F_{\theta,X}, Y)$ is $\beta$-sub-Gaussian with $\beta = \sqrt{2(\beta_1^2 + \beta_2^2)}$.*

*Proof.* We use following alternative representation of the CRPS, see Gneiting and Raftery (2007): for fixed $F \in \mathcal{P}_1(\mathbb{R})$ and $y \in \mathbb{R}$,

$$S(F, y) = \mathbb{E}[|Z - y|] - \frac{1}{2}\mathbb{E}[|Z' - Z|]$$

where $Z, Z'$ denote independent random variables with distribution $F$. The triangle inequality then implies

$$S(F, y) = \mathbb{E}[|Z - y|] - \frac{1}{2}\mathbb{E}[|Z' - Z|] \leq |y| + \frac{1}{2}\mathbb{E}[|Z + Z| - |Z' - Z|]$$
$$\leq |y| + \mathbb{E}[|Z|] = |y| + m_1(F),$$

with $m_1(F)$ the absolute moment of $F$. We deduce that, for all $\theta \in \Theta$,

$$S(F_{\theta,X}, Y) \leq |Y| + m_1(F_{\theta,X}) \tag{10}$$

where, by Assumption 1, $|Y|$ and $m_1(F_{\theta,X})$ are sub-Gaussian with parameter $\beta_1$ and $\beta_2$ respectively. Then, Proposition 3 implies that $S(F_{\theta,X}, Y)$ is $\beta$-sub-Gaussian with $\beta = \sqrt{2(\beta_1^2 + \beta_2^2)}$. $\square$

**Corollary 2.** *Under Assumption 1, for all $\theta \in \Theta$, the empirical risk $\hat{\mathcal{R}}_n(F_\theta)$ satisfies the concentration inequality*

$$\mathbb{P}\left(\left|\hat{\mathcal{R}}_n(F_\theta) - \mathcal{R}(F_\theta)\right| \geq t\right) \leq 2\exp\left(-\frac{nt^2}{2\beta^2}\right), \quad \text{for all } t \geq 0$$

*with $\beta = \sqrt{2(\beta_1^2 + \beta_2^2)}$.*

*Proof.* The random variables $Z_{\theta,i} = S(F_{\theta,X_i}, Y_i)$, $1 \leq i \leq n$, are i.i.d. with expectation

$$\mathbb{E}[Z_{\theta,i}] = \mathbb{E}[S(F_{\theta,X_i}, Y_i)] = \mathcal{R}(F_\theta)$$

and empirical mean

$$\frac{1}{n}\sum_{i=1}^n Z_{\theta,i} = \frac{1}{n}\sum_{i=1}^n S(F_{\theta,X_i}, Y_i) = \hat{\mathcal{R}}_n(F_\theta).$$

By Proposition 5, $Z_{\theta,i}$ is $\beta$-sub-Gaussian with $\beta = \sqrt{2(\beta_1^2 + \beta_2^2)}$. We can then apply the general Hoeffding's inequality (Proposition 4) and deduce the result. $\square$

## B  Proof of the results of Section 3.1

Before proving Theorem 1, we introduce preliminary results about the Lipschitz regularity of the CRPS and the empirical risk.

**Lemma 2.** *For all $F_1, F_2 \in \mathcal{P}_1(\mathbb{R})$ and $y \in \mathbb{R}$, we have*

$$|S(F_1, y) - S(F_2, y)| \leq 2W_1(F_1, F_2)$$

This Lemma states that the CRPS is 2-Lipschitz in the first variable with respect to the Wasserstein distance $W_1$.

*Proof.* By the definition (3) of the CRPS,

$$S(F_1, y) - S(F_2, y) = \int_{\mathbb{R}} \left((\mathbb{1}_{\{y \leq z\}} - F_1(z))^2 - (\mathbb{1}_{\{y \leq z\}} - F_2(z))^2\right) dz.$$

Using $a^2 - b^2 = (a-b)(a+b)$, we get

$$|S(F_1, y) - S(F_2, y)| \leq \int_{\mathbb{R}} |F_1(z) - F_2(z)||F_1(z) + F_2(z) - 2\mathbb{1}_{\{z \leq y\}}|dz$$

$$\leq 2 \int_{\mathbb{R}} |F_1(z) - F_2(z)|dz.$$

We recognize the expression of the Wasserstein distance of order 1 for probability measures on $\mathbb{R}$ as the $L^1$-distance between their cdf, i.e. the formula

$$\int_{\mathbb{R}} |F_1(z) - F_2(z)|dz = W_1(F_1, F_2),$$

see Bobkov and Ledoux (2019). The result follows. □

We can deduce that, under Assumption 2, the theoretical and empirical risks are also Lipschitz continuous.

**Proposition 6.** *Under Assumption 2, for all $\theta_1, \theta_2 \in \Theta$,*

$$|\mathcal{R}(F_{\theta_1}) - \mathcal{R}(F_{\theta_2})| \leq 2L\|\theta_1 - \theta_2\|$$

*and*

$$|\hat{\mathcal{R}}_n(F_{\theta_1}) - \hat{\mathcal{R}}_n(F_{\theta_2})| \leq 2L\|\theta_1 - \theta_2\|.$$

*Proof.* Using the definition of the theoretical risk and the Lipschitz continuity of the CRPS, we have

$$|\mathcal{R}(F_{\theta_1}) - \mathcal{R}(F_{\theta_2})| = |\mathbb{E}[S(F_{\theta_1, X}, Y) - S(F_{\theta_2, X}, Y)]|$$
$$\leq \mathbb{E}[|S(F_{\theta_1, X}, Y) - S(F_{\theta_2, X}, Y)|]$$
$$\leq 2\mathbb{E}[W_1(F_{\theta_1, X}, F_{\theta_2, X})].$$

Then, Assumption 2 implies

$$|\mathcal{R}(F_{\theta_1}) - \mathcal{R}(F_{\theta_2})\| \leq 2L\|\theta_1 - \theta_2\|.$$

Similarly for the empirical risk,

$$|\hat{\mathcal{R}}_n(F_{\theta_1}) - \hat{\mathcal{R}}_n(F_{\theta_2})| \leq \frac{1}{n} \sum_{i=1}^n \left| S(F_{\theta_1, X_i}, Y_i) - S(F_{\theta_2, X_i}, Y_i)] \right|$$

$$\leq \frac{2}{n} \sum_{i=1}^n W_1(F_{\theta_1, X_i}, F_{\theta_2, X_i})$$

$$\leq 2L\|\theta_1 - \theta_2\|.$$

□

We are now ready for the proof of Theorem 1.

*Proof of Theorem 1.* By compactness of $\Theta$ and continuity of $\theta \mapsto \mathcal{R}(\theta)$, the theoretical risk reaches a minimum on $\Theta$ and we can define $\theta^* = \arg\min_{\theta \in \Theta} \mathcal{R}(F_\theta)$. Similarly, $\hat{\theta}_n = \arg\min_{\theta \in \Theta} \hat{\mathcal{R}}_n(F_\theta)$ is well defined. The estimation error can then be decomposed into three terms

$$\mathcal{R}(F_{\hat{\theta}_n}) - \inf_{\theta \in \Theta} \mathcal{R}(F_\theta)$$

$$= \mathcal{R}(F_{\hat{\theta}_n}) - \mathcal{R}(F_{\theta^*})$$

$$= \left( \mathcal{R}(F_{\hat{\theta}_n}) - \hat{\mathcal{R}}_n(F_{\hat{\theta}_n}) \right) + \left( \hat{\mathcal{R}}_n(F_{\hat{\theta}_n}) - \hat{\mathcal{R}}_n(F_{\theta^*}) \right) + \left( \hat{\mathcal{R}}_n(F_{\theta^*}) - \mathcal{R}(F_{\theta^*}) \right) \quad (11)$$

By definition of $\hat{\theta}_n$, the second term is non-positive and we deduce

$$\left| \mathcal{R}(F_{\hat{\theta}_n}) - \inf_{\theta \in \Theta} \mathcal{R}(F_\theta) \right| \leq 2 \sup_{\theta \in \Theta} \left| \hat{\mathcal{R}}_n(F_\theta) - \mathcal{R}(F_\theta) \right|. \quad (12)$$

Since $\Theta$ is compact, it can be included in some centered closed Euclidean ball $\bar{B}_R$ with radius $R > 0$. Therefore, for all $\epsilon \leq R$, there exists an $\epsilon$-net $\Theta_\epsilon$ of $\Theta$ such that $\mathrm{card}(\Theta_\epsilon) \leq \left(\frac{3R}{\epsilon}\right)^K$, see Devroye et al. (1996). The term $\epsilon$-net means that, for all $\theta \in \Theta$, there exists $\theta_\epsilon \in \Theta_\epsilon$ such that $\|\theta - \theta_\epsilon\| \leq \epsilon$. Now, for all $\theta \in \Theta$, we introduce the decomposition

$$\left|\hat{\mathcal{R}}_n\left(F_\theta\right) - \mathcal{R}\left(F_\theta\right)\right| \leq \left|\hat{\mathcal{R}}_n\left(F_\theta\right) - \hat{\mathcal{R}}_n\left(F_{\theta_\epsilon}\right)\right| + \left|\hat{\mathcal{R}}_n\left(F_{\theta_\epsilon}\right) - \mathcal{R}\left(F_{\theta_\epsilon}\right)\right| + \left|\mathcal{R}\left(F_{\theta_\epsilon}\right) - \mathcal{R}\left(F_\theta\right)\right|.$$

By the Lipschitz properties stated in Proposition 6, the first and third terms are bounded from above by $2L\epsilon$. Therefore, we deduce

$$2\sup_{\theta\in\Theta}\left|\hat{\mathcal{R}}_n\left(F_\theta\right) - \mathcal{R}\left(F_\theta\right)\right| \leq 8L\epsilon + 2\sup_{\theta\in\Theta_\epsilon}\left|\hat{\mathcal{R}}_n\left(F_\theta\right) - \mathcal{R}\left(F_\theta\right)\right|, \tag{13}$$

which implies, for $t \geq 16L\epsilon$,

$$\mathbb{P}\left(2\sup_{\theta\in\Theta}\left|\hat{\mathcal{R}}_n\left(F_\theta\right) - \mathcal{R}\left(F_\theta\right)\right| \geq t\right) \leq \mathbb{P}\left(\sup_{\theta\in\Theta_\epsilon}\left|\hat{\mathcal{R}}_n\left(F_\theta\right) - \mathcal{R}\left(F_\theta\right)\right| \geq t/4\right).$$

We then use Corollary 2 stating that, for all $\theta \in \Theta$,

$$\mathbb{P}\left(\left|\hat{\mathcal{R}}_n(F_\theta) - \mathcal{R}(F_\theta)\right| \geq t\right) \leq 2\exp\left(-\frac{nt^2}{2\beta^2}\right)$$

with $\beta = \sqrt{2(\beta_1^2 + \beta_2^2)}$. The union bound and the inequality $\mathrm{card}(\Theta_\epsilon) \leq \left(\frac{3R}{\epsilon}\right)^K$ imply, for $t \geq 16L\epsilon$,

$$\mathbb{P}\left(2\sup_{\theta\in\Theta}\left|\hat{\mathcal{R}}_n\left(F_\theta\right) - \mathcal{R}\left(F_\theta\right)\right| \geq t\right) \leq \sum_{\theta\in\Theta_\epsilon}\mathbb{P}\left(\left|\hat{\mathcal{R}}_n\left(F_\theta\right) - \mathcal{R}\left(F_\theta\right)\right| \geq t/4\right)$$

$$\leq 2\left(\frac{3R}{\epsilon}\right)^K \exp\left(-\frac{nt^2}{32\beta^2}\right).$$

Setting $\epsilon = 3R/n$ in this inequality and using Eq. (12), we deduce

$$\mathbb{P}\left(\left|\mathcal{R}(F_{\hat{\theta}_n}) - \inf_{\theta\in\Theta}\mathcal{R}(F_\theta)\right| \geq t\right) \leq 2n^K \exp\left(-\frac{nt^2}{32\beta^2}\right), \quad t \geq \frac{48LR}{n}. \tag{14}$$

The right hand side is equal to $\delta$ for $t = \sqrt{32\beta^2 \log(2n^K/\delta)/n}$, yielding the result. Note that we need this specific choice of $t$ to satisfy $t \geq 48LR/n$, or equivalently $n\log(2n^K/\delta) \geq (48LR)^2/c_\beta$ with $c_\beta = 64(\beta_1^2 + \beta_2^2)$. $\qquad\square$

*Proof of Corollary 1.* Setting $a = \log(2n^K)$ and $b = n/c_\beta$, Eq. (14) can be rewritten as

$$\mathbb{P}\left(\left|\mathcal{R}(F_{\hat{\theta}_n}) - \inf_{\theta\in\Theta}\mathcal{R}(F_\theta)\right| \geq t\right) \leq \exp\left(a - bt^2\right), \quad t \geq 48LR/n.$$

For $t \leq \sqrt{a/b}$, the inequality is trivial because the right hand side is larger than 1. We deduce that the inequality holds for all $t \geq 0$ as soon as $48LR/n \leq \sqrt{a/b}$ which is equivalent to $n\log(2n^K) \geq (48LR)^2/c_\beta$, which holds for $n$ large enough. Then we can apply Lemma 1 and deduce the upper bound $2\sqrt{a/b}$ for the expectation of the estimation error. $\qquad\square$

## C  Proof of the results of Section 3.2

*Proof of Theorem 2.* A straightforward adaptation of the proof of Proposition 5 shows that Assumption 3 implies that the random variables $S(\hat{F}_{n,X}^m, Y)$, $m = 1, \ldots, M$, are $\beta$-sub-Gaussian conditionally on $\mathcal{D}_n$ with $\beta = \sqrt{2(\beta_1^2 + \beta_n^2)}$. Then, similarly as in Corollary 2, Hoeffding inequality implies

$$\mathbb{P}\left(\left|\hat{\mathcal{R}}_N'(\hat{F}_n^m) - \mathcal{R}(\hat{F}_n^m)\right| \geq t \,\Big|\, \mathcal{D}_n\right) \leq 2\exp\left(-\frac{Nt^2}{4(\beta_1^2 + \beta_n^2)}\right) \tag{15}$$

for all $t \geq 0$ and $m = 1, \ldots, M$. With a similar reasoning as in Eq. (11), the regret in model selection is bounded from above by

$$
\begin{aligned}
& \mathcal{R}(\hat{F}_n^{\hat{m}}) - \min_{1 \leq m \leq M} \mathcal{R}(\hat{F}_n^m) \\
&= \mathcal{R}(\hat{F}_n^{\hat{m}}) - \mathcal{R}(\hat{F}_n^{m^*}) \\
&= \left( \mathcal{R}(\hat{F}_n^{\hat{m}}) - \hat{\mathcal{R}}_N'(\hat{F}_n^{\hat{m}}) \right) + \left( \hat{\mathcal{R}}_N'(\hat{F}_n^{\hat{m}}) - \hat{\mathcal{R}}_N'(\hat{F}_n^{m^*}) \right) + \left( \hat{\mathcal{R}}_N'(\hat{F}_n^{m^*}) - \mathcal{R}(\hat{F}_n^{m^*}) \right) \\
&\leq 2 \max_{1 \leq m \leq M} \left| \hat{\mathcal{R}}_N'(\hat{F}_n^m) - \mathcal{R}(\hat{F}_n^m) \right|.
\end{aligned}
\tag{16}
$$

This together with the union bound and Eq. (15) implies

$$
\begin{aligned}
\mathbb{P} \left( \mathcal{R}(\hat{F}_n^{\hat{m}}) - \min_{1 \leq m \leq M} \mathcal{R}(\hat{F}_n^m) \geq t \,\Big|\, \mathcal{D}_n \right) &\leq \mathbb{P} \left( \max_{1 \leq m \leq M} \left| \hat{\mathcal{R}}_N'(\hat{F}_n^m) - \mathcal{R}(\hat{F}_n^m) \right| \geq \frac{t}{2} \,\Big|\, \mathcal{D}_n \right) \\
&\leq \sum_{m=1}^M \mathbb{P} \left( \left| \hat{\mathcal{R}}_N'(\hat{F}_n^m) - \mathcal{R}(\hat{F}_n^m) \right| \geq \frac{t}{2} \,\Big|\, \mathcal{D}_n \right) \\
&\leq 2M \exp \left( -\frac{Nt^2}{16(\beta_1^2 + \beta_n^2)} \right).
\end{aligned}
$$

The right hand side is equal to $\delta$ for $t = 4\sqrt{c_n \log(2M/\delta)/N}$, which proves the first claim. The second claim follows by an application of Lemma 1 with $a = \log(2M)$ and $b = N/(16c_n)$. $\qquad\square$

For the proof of Theorem 3, we need the following Lemma.

**Lemma 3.** *Consider cumulative distribution functions* $(G^m)_{1 \leq m \leq M}$ *and, for* $\lambda \in \Lambda$, *the convex aggregation* $G^\lambda = \sum_{m=1}^M \lambda_m G^m$. *Then the following Lipschitz property is satisfied: for all* $\lambda_1, \lambda_2 \in \Lambda$,

$$
W_1(G^{\lambda_1}, G^{\lambda_2}) \leq \max_{1 \leq m \leq M} m_1(G^m) \|\lambda_1 - \lambda_2\|_1
$$

*Proof.* In dimension 1, the Wasserstein distance is given by

$$
W_1(G^{\lambda_1}, G^{\lambda_2}) = \int_{\mathbb{R}} |G^{\lambda_1}(y) - G^{\lambda_2}(y)| \mathrm{d}y.
$$

By definition of $G^\lambda$, we deduce

$$
\begin{aligned}
W_1(G^{\lambda_1}, G^{\lambda_2}) &= \int_{\mathbb{R}} \Big| \sum_{m=1}^M (\lambda_{1,m} - \lambda_{2,m})(G^m(y) - \mathbb{1}_{\{y \geq 0\}}) \Big| \mathrm{d}y \\
&\leq \sum_{m=1}^M |\lambda_{1,m} - \lambda_{2,m}| \int_{\mathbb{R}} |G^m(y) - \mathbb{1}_{\{y \geq 0\}}| \mathrm{d}y \\
&= \sum_{m=1}^M |\lambda_{1,m} - \lambda_{2,m}| \, m_1(G^m) \\
&\leq \max_{1 \leq m \leq M} m_1(G^m) \|\lambda_1 - \lambda_2\|_1.
\end{aligned}
$$

In the first line, we use that $\sum_{m=1}^M (\lambda_{1,m} - \lambda_{2,m}) = 0$ so that we can introduce the indicator function $\mathbb{1}_{\{y \geq 0\}}$. In the third line, we use $\int_{\mathbb{R}} |G^m(y) - \mathbb{1}_{\{y \geq 0\}}| \mathrm{d}y = W_1(G^m, \delta_0) = m_1(G^m)$. $\qquad\square$

*Proof of Theorem 3.* We work conditionally on $\mathcal{D}_n$ so that $\hat{F}_n^m$, $1 \leq m \leq M$, can be seen as deterministic. The empirical risk is computed on the validation set $\mathcal{D}_N'$ with cardinal $N$. Furthermore, Assumption 3 implies that, conditionally on $\mathcal{D}_n$, $Y$ and $\hat{F}_n^\lambda$ satisfy Assumption 1 with $\Theta$ replaced by $\Lambda$ and constant $\beta_2$ replaced by $\beta_n$. Lemma 3 implies that, conditionally on $\mathcal{D}_n$, $F_x^\lambda$ satisfy Assumption 2 with $\Theta$ replaced by $\Lambda_M$ and constant $L$ replaced by $\sqrt{M} \max_{1 \leq m \leq M} m_1(\hat{F}_{n,x}^m)$. As a consequence, we can apply Theorem 1 and deduce the result. $\qquad\square$

# D  Proofs and additional results for Section 3.3

## D.1  Proof of Theorem 4

Without sub-Gaussian assumptions, we cannot use Hoeffding inequality. Alternatively, we use a consequence of Rosenthal inequality providing a simple control of moments. For a reference, see Petrov (1995, Theorems 2.9 and 2.10).

**Proposition 7.** *Let $Z, Z_1, \ldots, Z_n$ be i.i.d. random variables such that $m_p(Z) = \mathbb{E}[|Z - \mathbb{E}[Z]|^p] < \infty$ for some $p \geq 2$. Then,*

$$\mathbb{E}\left[\left|\frac{1}{n}\sum_{i=1}^{n}\left(Z_i - \mathbb{E}[Z_i]\right)\right|^p\right] \leq c(p)m_p(Z)n^{-p/2},$$

*with $c(p) > 0$ depending only on $p$.*

We deduce a simple moment bound for the empirical risk.

**Proposition 8.** *Under Assumption 4, we have, for all $\theta \in \Theta$*

$$\mathbb{E}\left[\left|\hat{\mathcal{R}}_n(F_\theta) - \mathcal{R}(F_\theta)\right|^p\right] \leq c'(p)Dn^{-p/2}$$

*with $c'(p) = 4^p c(p)$ depending only on $p$.*

*Proof.* Eq. (10) together with the upper bound $|a + b|^p \leq 2^{p-1}(|a|^p + |b|^p)$ imply

$$|S(F_{\theta,X}, Y)|^p \leq \left(|Y| + m_1(F_{\theta,X})\right)^p \leq 2^{p-1}\left(|Y|^p + m_1(F_{\theta,X})^p\right).$$

Taking the expectation and using Assumption 4, we deduce, with the notation $Z_\theta = S(F_{\theta,X}, Y)$,

$$\mathbb{E}[|Z_\theta|^p] \leq 2^{p-1}\left(|Y|^p + m_1(F_{\theta,X})^p\right) \leq 2^p D.$$

By Jensen inequality, this implies the upper bound

$$\mathbb{E}\left[|Z_\theta - \mathbb{E}[Z_\theta]|^p\right] \leq 4^p D.$$

The centered empirical risk can be written as

$$\hat{\mathcal{R}}_n(F_\theta) - \mathcal{R}(F_\theta) = \frac{1}{n}\sum_{i=1}^{n}\left(Z_{\theta,i} - \mathbb{E}[Z_{\theta,i}]\right)$$

with $Z_{\theta,i} = S(F_{\theta,X_i}, Y_i)$, $1 \leq i \leq n$, i.i.d. random variables. Then the result follows from an application of Proposition 7. □

*Proof of Theorem 4.* We use the same notation as in the proof of Theorem 1. The beginning of the proof follows exactly the same lines: according to Eq. 12-13, for all $\epsilon$-net $\Theta_\epsilon \subset \Theta$, we have

$$\left|\mathcal{R}(F_{\hat{\theta}_n}) - \inf_{\theta \in \Theta}\mathcal{R}(F_\theta)\right| \leq 8L\epsilon + 2\sup_{\theta \in \Theta_\epsilon}\left|\hat{\mathcal{R}}_n(F_\theta) - \mathcal{R}(F_\theta)\right|. \tag{17}$$

This implies

$$\mathbb{E}\left[\left|\mathcal{R}(F_{\hat{\theta}_n}) - \inf_{\theta \in \Theta}\mathcal{R}(F_\theta)\right|^p\right] \leq 2^{p-1}\left((8L\epsilon)^p + 2^p\mathbb{E}\left[\sup_{\theta \in \Theta_\epsilon}\left|\hat{\mathcal{R}}_n(F_\theta) - \mathcal{R}(F_\theta)\right|^p\right]\right).$$

The expectation in the right hand side is bounded from above by

$$\mathbb{E}\left[\sup_{\theta \in \Theta_\epsilon}\left|\hat{\mathcal{R}}_n(F_\theta) - \mathcal{R}(F_\theta)\right|^p\right] \leq \sum_{\theta \in \Theta_\epsilon}\mathbb{E}\left[\left|\hat{\mathcal{R}}_n(F_\theta) - \mathcal{R}(F_\theta)\right|^p\right]$$

$$\leq (3R)^K\epsilon^{-K}c'(p)Dn^{-p/2}$$

where the last inequality follows from Proposition 8 and the bound $\mathrm{card}(\Theta_\epsilon) \leq (3R/\epsilon)^K$. Hence we deduce

$$\mathbb{E}\left[\left|\mathcal{R}(F_{\hat{\theta}_n}) - \inf_{\theta \in \Theta}\mathcal{R}(F_\theta)\right|^p\right] \leq 2^{p-1}\left((8L\epsilon)^p + 2^p(3R)^K\epsilon^{-K}c'(p)Dn^{-p/2}\right).$$

Minimizing the right hand side with respect to $\epsilon$ according to Lemma 4 below and taking the $p$-th root, we obtain

$$\mathbb{E}\left[\left|\mathcal{R}(F_{\hat{\theta}_n}) - \inf_{\theta \in \Theta} \mathcal{R}(F_\theta)\right|^p\right]^{1/p} \leq Cn^{-p/(2(p+K))}$$

where the constant $C = C(K, p, L, D, R)$ does not depend on $n$ and can be made explicit thanks to Lemma 4. Finally, Jensen's inequality yields the result. $\quad\square$

**Lemma 4.** *For all $a, b > 0$ and $p, q > 0$,*

$$\inf_{\epsilon > 0} \left(a\epsilon^p + b\epsilon^{-q}\right) = C_{p,q} a^{\frac{q}{p+q}} b^{\frac{p}{p+q}},$$

*with $C_{p,q} = \left(\frac{p}{q}\right)^{\frac{q}{p+q}} + \left(\frac{q}{p}\right)^{\frac{p}{p+q}}$.*

*Proof.* A straightforward analysis of the function $\epsilon \mapsto a\epsilon^p + b\epsilon^{-q}$ shows that its derivative vanishes at $\epsilon = \left(\frac{bq}{ap}\right)^{\frac{1}{p+q}}$ where the minimum is reached. $\quad\square$

### D.2 Additional results

We provide some additional results to Theorem 4 from Section 3.3 regarding model selection and convex aggregation. Our working assumption in this framework is the following.

**Assumption 5.** *For $p \geq 2$,*

  *i)* $\|Y\|_{L^p}^p \leq D$;

  *ii)* *conditionnaly on $\mathcal{D}_n$, $\|m_1(\hat{F}_{n,X}^m)\|_{L^p}^p \leq D_n = D(\mathcal{D}_n)$ for all $m = 1, \ldots, M$.*

For instance, one can show with a reasoning similar to Proposition 1 that for the EMOS and DRN models, Assumption 5 holds as soon as $Y$ and $F$ have a finite moment of order $p$ and $X$ remains bounded. For the KNN and DRF models, a reasoning similar to Proposition 2 shows that the assumption holds with $D_n = \max_{1 \leq i \leq n} |Y_i|^p$.

**Model selection.** Here is our result on model selection under moment assumption only.

**Theorem 5.** *Under Assumption 5, we have*

$$\mathbb{E}\left[\mathcal{R}(\hat{F}_n^{\hat{m}}) - \min_{1 \leq m \leq M} \mathcal{R}(\hat{F}_n^m) \Big| \mathcal{D}_n\right] \leq 2\big(c'(p)\max(D, D_n)M\big)^{1/p} N^{-1/2},$$

*with $c'(p)$ the constant from Proposition 8.*

Before proving the Theorem, we establish the following result.

**Proposition 9.** *Under Assumption 5, we have, for all $1 \leq m \leq M$,*

$$\mathbb{E}\left[\left|\hat{\mathcal{R}}_N'(\hat{F}_n^m) - \mathcal{R}(\hat{F}_n^m)\right|^p \Big| \mathcal{D}_n\right] \leq c'(p)\max(D, D_n)N^{-p/2}.$$

*Proof.* We work conditionally on $\mathcal{D}_n$ so that $\hat{F}_n^m$ can be seen as a deterministic quantity. The empirical risk is computed on the validation set $\mathcal{D}_N'$ with cardinal $N$. Furthermore, Assumption 5 implies that, conditionally on $\mathcal{D}_n$, $Y$ and $\hat{F}_n^m$ satisfy 4 with constant $D$ replaced by $\max(D, D_n)$. Given these remarks, Proposition 9 follows by an application of Proposition 8 where $F_\theta$ is replaced by $\hat{F}_n^m$, $\hat{\mathcal{R}}_n$ is replaced by $\hat{\mathcal{R}}_N'$ and the expectation by the conditional expectation given $\mathcal{D}_n$. $\quad\square$

*Proof of Theorem 5.* Thanks to Eq. (16),

$$\mathcal{R}(\hat{F}_n^{\hat{m}}) - \min_{1 \leq m \leq M} \mathcal{R}(\hat{F}_n^m) \leq 2 \max_{1 \leq m \leq M} \left|\hat{\mathcal{R}}_N'(\hat{F}_n^m) - \mathcal{R}(\hat{F}_n^m)\right|,$$

whence we deduce

$$\mathbb{E}\left[\left|\mathcal{R}(\hat{F}_n^{\hat{m}}) - \min_{1 \le m \le M} \mathcal{R}(\hat{F}_n^m)\right|^p \Big| \mathcal{D}_n\right] \le 2^p \mathbb{E}\left[\max_{1 \le m \le M}\left|\hat{\mathcal{R}}_N'(\hat{F}_n^m) - \mathcal{R}(\hat{F}_n^m)\right|^p \Big| \mathcal{D}_n\right]$$

$$\le 2^p \sum_{m=1}^{M} \mathbb{E}\left[\left|\hat{\mathcal{R}}_N'(\hat{F}_n^m) - \mathcal{R}(\hat{F}_n^m)\right|^p \Big| \mathcal{D}_n\right]$$

$$\le 2^p M c'(p) \max(D, D_n) N^{-p/2}.$$

The result follows by Jensen's inequality. □

**Convex aggregation.** Here is our main result on convex aggregation under moment assumption only.

**Theorem 6.** *Under Assumption 5, we have*

$$\mathbb{E}\left[\mathcal{R}(\hat{F}_n^{\hat{\lambda}}) - \inf_{\lambda \in \Lambda_M} \mathcal{R}(\hat{F}_n^\lambda) \Big| \mathcal{D}_n\right] \le C\left(L^K \max(D, D_n) N^{-p/2}\right)^{1/p+K}$$

*with $L = \sqrt{M} \max_{1 \le m \le M} m_1(\hat{F}_n^m)$ and constant $C > 0$ depending only on $p$ and $K$.*

*Proof of Theorem 6.* The proof follows by an application of Theorem 4 exactly in the same way that Theorem 3 was derived from Theorem 1. □

# E  Proofs for Section 4

The following Definition and Lemma about location/scale families are useful for the proof of Section 4.

**Definition 3.** *Let $F$ be a probability distribution on $\mathbb{R}$ with zero mean and unit variance. The associated location-scale family is defined as $\{F_{m,\sigma}, \ m \in \mathbb{R}, \ \sigma > 0\}$ where $F_{m,\sigma}$ is the law of $m + \sigma Z$ with $Z \sim F$.*

**Proposition 10** (Chhachhi and Teng 2023). *For all $m_1, m_2 \in \mathbb{R}$ and $\sigma_1, \sigma_2 > 0$, we have*

$$W_1(F_{m_1,\sigma_1}, F_{m_2,\sigma_2}) \le |m_1 - m_2| + m_1(F) |\sigma_1 - \sigma_2|.$$

*Proof of Proposition 1.* Denote by $F_{m(x;\theta),\sigma(x;\theta)}$ the output distribution of the DRN with parameter $\theta \in \Theta$ and input $x$, where $m(x;\theta)$ and $\sigma(x;\theta)$ are the location and scale parameters given by Eq. (7). Because the activation function $g$ is assumed Lipschitz continuous, when $x$ and $\theta$ remain in bounded sets, the functions $\theta \mapsto m(x;\theta)$ and $\theta \mapsto \sigma(x;\theta)$ are Lipschitz continuous with a Lipschitz constant bounded by some constant $C$ uniformly in $x$. Then, Proposition 10 implies that the output distributions satisfy

$$W_1(F_{m(x;\theta_1),\sigma(x;\theta_1)}, F_{m(x;\theta_2),\sigma(x;\theta_2)}) \le |m(x;\theta_1) - m(x;\theta_2)| + m_1(F) |\sigma(x;\theta_1) - \sigma(x;\theta_2)|$$

$$\le C(1 + m_1(F))\|\theta_1 - \theta_2\|.$$

for all $\theta$ in $\Theta$ compact and $x$ in the support of $X$ bounded. It follows that Assumption 2 is satisfied. Furthermore, this estimation of the Wasserstein distance implies that the absolute moment $m_1(F_{\theta,X})$ remains bounded for all $\theta \in \Theta$ and $x$ in the support of $X$. Hence $m_1(F_{\theta,X})$ is sub-Gaussian (uniformly in $\theta$) and this proves Assumption 1. For the same reason, the absolute moment $m_1(\hat{F}_n) = m_1(F_{\hat{\theta}_n,X})$ remains bounded and this implies Assumption 3 with $\beta(\mathcal{D}_n)$ constant not depending on the training set. □

*Proof of Proposition 2.* The two models KNN and DRF can be written in the form

$$\hat{F}_{n,x}(y) = \sum_{i=1}^{n} w_{ni}(x)\mathbb{1}_{\{Y_i \le y\}}$$

for suitable probability weights $(w_{ni}(x))_{1 \le i \le n}$. Hence the absolute moment of the predictive distribution satisfies

$$m_1(\hat{F}_{n,x}) = \sum_{i=1}^{n} w_{ni}(x)|Y_i| \le \max_{1 \le i \le n} |Y_i|.$$

We can see that the random variable $m_1(\hat{F}_{n,X})$ remains bounded and is hence $\beta_n$-sub-Gaussian with $\beta_n = \max_{1 \le i \le n} |Y_i|$, which proves Assumption 3. □

# F   Additional numerical results

The numerical results for the `airfoil` data is presented in Figure 2 and Table 2. The left plot of Figure 2 corresponds to KNN, and the middle plot to DRF. In both instances, the validation curve exhibits a clear minimum, enabling the selection parameter values $\widehat{k} = 3$ and $\widehat{\mathtt{mtry}} = 5$ associated with CRPS-error of 2.16 and 1.47 on the validation set respectively. Then the models KNN, DRF, MS and CA are tested on the test set, and the right plot of Figure 2 shows the test error evaluated with 100 repetitions. The boxplots reveal a clear difference between the test errors for KNN and DRF in favor of DRF. In this case, MS achieves the same performance as DRF (in fact DRF is systematically selected), and a slight improve is achieved with CA. These results are confirmed by the numerical values in Table 2.

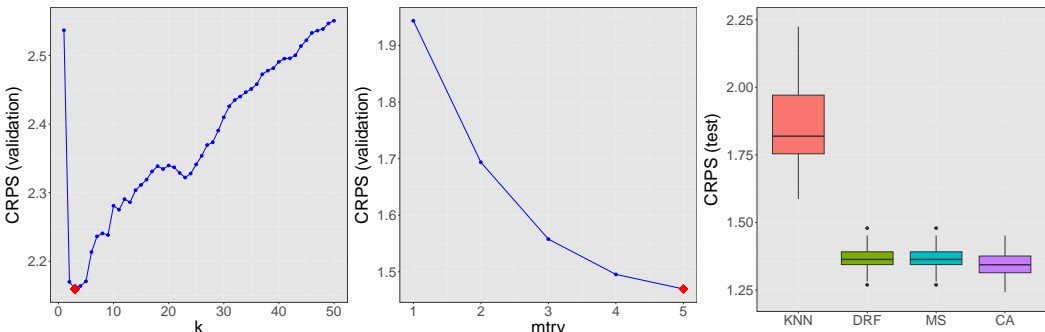

Figure 2: `airfoil` data. Left and middle: selection of `k` for the KNN algorithm and of `mtry` for the DRF algorithm by minimization of the validation error. Right: test error evaluated with 100 repetitions for KNN, DRF, model selection (MS) and convex aggregation (CA).

| KNN | DRF | MS | CA |
|---|---|---|---|
| 1.864 (0.015) | 1.366 (0.004) | 1.366 (0.004) | 1.344 (0.005) |

Table 2: `airfoil` data. Mean of the test CRPS and its standard error (in parenthesis) over 100 repetitions.

