# OpenReview forum: "Distributional regression: CRPS-error bounds for model fitting, model selection and convex aggregation"
_NeurIPS.cc/2024/Conference — NeurIPS 2024 poster_

### Official Review · Reviewer_kUjo · 2024-07-10

**Soundness:** 2
**Presentation:** 2
**Contribution:** 3
**Rating:** 5
**Confidence:** 4

**Summary:**

The paper studies distributional regression, a statistical technique that models not just the conditional mean of the response variable given the predictors (as in traditional regression) but the entire conditional distribution. The learning task minimizes the risk function, measured by the Continuous Rank Probability Score (CRPS). The authors aim to estimate the error bounds for the expectation of the predictive distribution. They also investigate model selection and aggregation via CRPS minimization on a validation set.

**Strengths:**

* Comprehensive Approach: The use of distributional regression provides a more holistic view of the data by modeling the entire conditional distribution, offering more insights compared to traditional regression methods. This work explores various aspects of this model and provides a thorough analysis.

* Theoretical Contributions: The authors present a robust theoretical framework, including the estimation of error bounds for the expectation of the predictive distribution. This adds valuable understanding and reliability to the model's predictions.

* Model Selection and Aggregation: The paper addresses practical aspects such as model selection and aggregation using CRPS minimization on a validation set, which is crucial for effectively applying the methodology in real-world scenarios.

**Weaknesses:**

* Typos in the Text: The text should be checked again by the authors to remove the typos (e.g., line 88: "on can show" should be "one can show").

* Math Symbols and Abbreviations: Several math symbols or abbreviations are used in the text without prior definition, reducing the readability of the text. For instance, "EMOS" in the abstract or $\mathcal{G}$ in line 69. Additionally, some symbols are used multiple times with different meanings. For example, $\mathcal{D}$ in line 19 indicates the training set, while $\mathcal{D}$ in line 71 denotes a subset of the set of all probability measures on $\mathcal{R}$. It would be better to use symbols more carefully and ensure consistency throughout the text.

* Comparison with Existing Methods: While the paper focuses on the advantages of the proposed CRPS-error bounds for distributional regression, it would be beneficial to include a comparison with existing techniques, both in terms of theoretical properties and empirical performance. Related works and available baselines have not been properly discussed in the paper. Including such comparisons would provide a clearer context for the contributions of the proposed method and its relative performance.

* Complexity Analysis: A detailed complexity analysis is needed to understand the computational cost of the proposed method, especially in comparison with other techniques.


* Numerical Analysis Clarity: The results presented in the numerical analysis should be clearer, with a more detailed explanation of what the numbers represent and how they support the claims made in the paper. The numerical analysis is not sufficient, and not enough sensitivity analysis has been provided. Only two small datasets have been used, and the simulation does not cover all aspects of the model. It would be better if the authors designed new experiments with different datasets and settings to comprehensively validate the proposed method. This would help demonstrate the robustness and applicability of the approach across various scenarios.

**Questions:**

See Weaknesses.

**Limitations:**

The limitations have not been properly discussed in the paper.

---

> ### Author Rebuttal · Authors · 2024-08-02
>
> Thank you for your work and time for reviewing the paper. We believe indeed that distributional regression is an important technique providing a more holistic view of the data and we stress that, despite its importance in practice, CRPS minimization in distributional regression has been very little investigated from a theoretical point of view. Please find below answers to your questions and concerns.
>
> 1) We have run again a spell checker and corrected a few typos.
> 2) Following your advice, we have used a different notation for $\mathcal{D}$ (replaced by $\mathcal{P}_0$) in order to avoid confusion with the training set $\mathcal{D}_n$. We have also checked that abbreviations are correctly explained when introduced for the first time.
> 3) We have chosen to focus in this work on the CRPS minimization technique because it is the most widely used in applications of distributional regression (see the references in the introduction), in particular in the context of ensemble forecast where the log-score and likelihood methods are not available. Despite it common application, there is a shortage of theoretical results regarding CRPS minimization for distributional regression which is the main motivation for the paper. Comparison with existing methods (e.g. minimization of other scoring rules) is left for further research.
> 4) Strictly speaking, we do not propose a new method but rather provide solid theoretical ground for existing methods. The original part of the paper is thus the concentration bound establishing the consistency of commonly used method in distributional regression. This is why our paper does not provide a detailed numerical analysis (only an illustration) nor a complexity analysis, but rather focuses on the theoretical aspects.

---

> > ### Comment · Reviewer_kUjo · 2024-08-12
> > **Response to the authors' feedback**
> >
> > Thank you for your revisions and the effort you’ve put into improving the manuscript. The gap between theoretical claims and numerical experiments still exists. After careful consideration, I have decided to retain my original score.

---

### Official Review · Reviewer_TyYn · 2024-07-12

**Soundness:** 2
**Presentation:** 3
**Contribution:** 2
**Rating:** 4
**Confidence:** 4

**Summary:**

This paper considers the problem of conditional distribution prediction. For covariate-response pair $(X, Y)$, the objective is to estimate the conditional distribution of $Y|X=x$ for all $x$. The paper provides concentration bounds for the empirical risk minimization (ERM) estimator with continuous rank probability score (CRPS) as loss. It also provides regret bounds for the model selection and model aggregation with CRPS minimization.

**Strengths:**

This paper is well-structured and clearly written. The proofs are rigorous, and the notations are well-organized. It offers theoretical guarantees that will benefit future research involving the CRPS minimization technique.

**Weaknesses:**

1. The proving technique for the theorems presented in the paper is pretty standard. The work lacks technical contribution.
2. In Theorem 1, the definition of $R$ which denotes the boundary of the parameter space $\Theta$ should be added in the main text.
3. The theorems have not provided insights for the algorithm design. The paper has not proposed new algorithms, but in line 262 it says "demonstrate the effectiveness of our proposed methods".
4. The entire experimental section does not utilize the theoretical guarantees established earlier. Additionally, there is a lack of experiments to validate the effectiveness and tightness of the theoretical bounds.

**Questions:**

As given in the "Weakness" section.

**Limitations:**

As given in the "Weakness" section.

---

> ### Author Rebuttal · Authors · 2024-08-02
>
> Thank you for overall positive appreciation of the paper! Please find below answers to your questions and concerns.
>
> Weaknesses:
> We agree that the technical development based on Hoeffding inequality is quite standard. Still, the paper provide the first detailed analysis of the CRPS risk for distributional regression and original contributions appear in this direction: Proposition 5, Lemmas 2 and 3 provide original results for the analysis of the CRPS, Propositions 1 and 2 provide original results for the analysis of various models (distributional nearest neighbours, EMOS, distributional neural networks). Our point of view is that the value of the work is to apply (standard) concentration techniques in an original context and to provide new insight into distributional regression, which is a fundamental task in statistics and machine learning.

---

> > ### Comment · Reviewer_TyYn · 2024-08-12
> >
> > Thank you very much for your detailed response. I will keep my original score.

---

### Official Review · Reviewer_uh9Z · 2024-07-13

**Soundness:** 3
**Presentation:** 3
**Contribution:** 2
**Rating:** 6
**Confidence:** 3

**Summary:**

The paper considers the problem of distributional regression, i.e., learning the distribution of Y conditional on X. In particular, the distributional regression problem is formulated as an empirical risk minimization problem, where standard concentration techniques are applied to obtain non-asymptotic bounds on the excessive risks.

**Strengths:**

1. The paper is well-written and easy to follow.
2. The question is well-motivated, and the technical development is solid.

**Weaknesses:**

After formulating the problem as an empirical risk minimization problem, applying concentration techniques appears to be very standard --- the technical contribution of the current paper is relatively weak.

**Questions:**

The current paper mainly considers the case of iid data, and correspondingly leverages concentration tools for iid data. I wonder if it could be generalized to adaptively collected data, since there are parallel concentration tools for such data.

**Limitations:**

The paper has discussed its limitation.

---

> ### Author Rebuttal · Authors · 2024-08-02
>
> Thank you for overall positive appreciation of the paper! Please find below answers to your questions and concerns.
>
> Weaknesses:
> We agree that the technical development based on Hoeffding inequality is quite standard. Still, the paper provide the first detailed analysis of the CRPS risk for distributional regression and original contributions appear in this direction: Proposition 5, Lemmas 2 and 3 provide original results for the analysis of the CRPS, Propositions 1 and 2 provide original results for the analysis of various models (distributional nearest neighbours, EMOS, distributional neural networks). Our point of view is that the value of the work is to apply (standard) concentration techniques in an original context and to provide new insight into distributional regression, which is a fundamental task in statistics and machine learning.

---

> > ### Comment · Reviewer_uh9Z · 2024-08-13
> >
> > Thank you for the response. I will remain my initial score.

---

### Official Review · Reviewer_GnPM · 2024-07-19

**Soundness:** 3
**Presentation:** 4
**Contribution:** 3
**Rating:** 6
**Confidence:** 3

**Summary:**

The paper provides theoretical guarantees for distributional regression, which aims to estimate the conditional distribution of a target random variable Y given covariates X.    These theoretical guarantees hold when the regression is learned by minimizing a particular proper scoring rule, the Continuous Rank Probability Score (CRPS), in the presence of i.i.d. data (X,Y).    Concentration bounds on the estimation error are provided for model fitting, model selection, and model averaging.

**Strengths:**

1.) The paper gives perhaps the clearest presentation I've ever seen when reviewing for one of the top ML conferences.   Its acceptance is almost warranted from this fact alone; there is significance in the exceedingly clear overview of distributional regression.

2.)  The provided concentration bounds cover multiple modeling goals (model fitting, model selection, and model averaging) and a wide range of popular models for distributional regression (both parametric and non-parametric).

**Weaknesses:**

1.) The experiments do not seem strongly related to the theory, other than that the experiments investigate distributional regression.  I expected the theoretical results to play a stronger role here.   Is there some way in which the theory guides the data analysis?  If not, what is the purpose of including an experiment at all? If nothing else, the experiments would seem to provide an opportunity to perform sanity checks.  For example, can the authors provide empirical evidence of consistency to the theoretical limiting values?

2.) Generalizing #1, the authors do not provide guidance on what their theoretical results can be used for.

**Questions:**

1.) The authors write: "[...] distributional regression is way stronger and more challenging than standard regression, where only a point prediction for Y given X=x is provided which typically reduces to an estimation of the conditional expectation."  I found this statement a bit puzzling.    For instance, both frequentist and Bayesian GLM's would seem to estimate conditional distributions of a target given explanatory variables, rather than simply providing point estimates.    Could the authors clarify?

Editing notes:

1.) "Way stronger and more challenging" (line 20) is redundant, and the first phrase is too colloquial.  Streamline to "more challenging" or perhaps "much more challenging".

2.) It is unclear why "we will mostly use the CRPS for discrete predictive distributions" (line 87).  Please clarify.

3.) Can the Wasserstein space `P_1(\R)` (defined between lines 83 and 84) be equivalently described as the set of probability distributions over `\R` with finite first moment? If so, this might be worth stating explicitly.

**Limitations:**

No limitations were provided.

---

> ### Author Rebuttal · Authors · 2024-08-02
>
> Thank you very much for the positive assessment of the quality of the paper presentation! Please find below answers to your questions and concerns.
>
> Weaknesses:
> 1) We agree that the experiment is loosely related to the theoretical results; our aim is to illustrate that the methods of interest (model selection, model aggregation) work well in practice; ideally it would be interesting to illustrate the convergence to zero of the estimation error, but we have to take into account that the theoretical limit value for the error (oracle) is unknown in our setting.
> 2) The main message is that the empirical risk minimization widely used in practice (be it for model fitting or model selection) is mathematically grounded. We have clarified and strengthened this message in the sections "Numerical Experiments" and "Conclusion". See in particular the sentence "These new theoretical results are solid mathematical justifications for common practices in the framework of distributional regression".
>
> Questions:
> 1) Indeed, generalized linear models can be seen as distributional regression techniques since the conditional distribution of Y given X=x is modelled by a distribution belonging to an exponential family.
> 2) Editing notes have been taken into account in the revised version.

---

> > ### Comment · Reviewer_GnPM · 2024-08-13
> >
> > Thank you for your response.  My initial assessment of strengths and weaknesses persists after the rebuttal.  Therefore, I maintain my original score.

---

### Author Rebuttal · Authors · 2024-08-02

We acknowledge the referees for their work and time spent on the paper. Overall, the comments are positive but two main concerns emerge from the reports that we comment below. All other (minor) suggestions have been taken into account.

1) Technical contributions --
We agree that the technical developments and concentration inequalities based on Hoeffding inequality are quite standard. Still, the paper provide the first detailed analysis of the CRPS risk for distributional regression and original contributions appear in this direction: Proposition 5, Lemmas 2 and 3 provide original results for the analysis of the CRPS, Propositions 1 and 2 provide original results for the analysis of various models (distributional nearest neighbours, EMOS, distributional neural networks). Our point of view is that the value of the work is to apply (standard) concentration techniques in an original context and to provide new insight into distributional regression, which is a fundamental task in statistics and machine learning. We believe the work will benefit future research involving CRPS minimization techniques in distributional regression.

2) Numerical experiments --
We agree that the experiment is loosely related to the theoretical results; our aim is to illustrate that the methods of interest (model selection, model aggregation) work well in practice; ideally it would be interesting to illustrate the convergence to zero of the estimation error, but we have to take into account that the theoretical limit value for the error (oracle) is unknown in our setting.

We very much hope that the reviewers will appreciate this point of view and support the work for presentation in Neurips.

---

> ### Comment · Area_Chair_5ZgV · 2024-08-12
> **Further comments by the reviewers?**
>
> Dear all reviewers:
>
> Can you please respond to the rebuttal as soon as possible? Your comments will be greatly appreciated. Many thanks,
>
> AC

---

### Decision · Program_Chairs · 2024-09-25

**Decision:**

Accept (poster)

**Comment:**

The paper provides concentration inequalities for the empirical risk minimization (ERM) estimator with continuous rank probability score (CRPS) as loss in the context of distributional regression, which aims to estimate the conditional distribution of a target random variable $Y$ given covariates $X$. Concentration bounds on the estimation error are provided for model fitting, model selection, and model averaging. The proof techniques are standard but the paper contains interesting results. The authors argue that the main takeaway of their theoretical results is that "the empirical risk minimization widely used in practice (be it for model fitting or model selection) is mathematically grounded". However, it might improve the paper substantially if the author(s) could use the theoretical results to provide deeper insights for the ERM methods.